



# New estimates of sulfate diffusion rates in the EPICA Dome C ice core

Rachael H. Rhodes[1], Yvan Bollet-Quivogne[1], Piers Barnes[2], Mirko Severi[3], Eric W. Wolff[1]

[1] - Department of Earth Sciences, University of Cambridge, UK
[2] - Imperial College London, Department of Physics, London, UK
[3] - Chemistry Department, University of Florence, Sesto F.no (FI) 50019, Italy

*Correspondence to*: Rachael H Rhodes (rhr34@cam.ac.uk)

**Abstract.** To extract climatically relevant chemical signals from the deepest, oldest Antarctic ice, we must first understand the degree to which chemical ions diffuse within solid ice. Volcanic sulfate peaks are the ideal target for such an investigation because they are high amplitude, short duration (~3 years) events with a quasi-uniform structure. Here we present analysis of

the EPICA Dome C sulfate record over the last 450 kyr. We identify volcanic peaks and isolate them from the non-sea salt sulfate background to reveal the effects of diffusion: amplitude damping and broadening of peaks in the time domain with increasing depth/age. Sulfate peak shape is also altered by the thinning of ice layers with depth that results from ice flow. Both processes must be simulated to derive effective diffusion rates. This is achieved by running a forward model to diffuse idealised sulfate peaks at different rates while also accounting for ice thinning. Our simulations suggest a median effective diffusion

rate of sulfate ions of $2.4 \pm 1.7 \times 10^{-7}$ m$^2$yr$^{-1}$ in the Holocene ice, slightly faster than suggested by previous work. The effective diffusion rate observed in deeper ice is significantly lower, and the Holocene ice shows the highest rate of the last 450 kyr. Beyond the Holocene, there is no systematic difference between the effective diffusion rates of glacial and interglacial periods despite variations in soluble ions concentrations, dust loading and ice grain radii. Effective diffusion rates for 40 to 200 ka are relatively constant, on the order of $1 \times 10^{-8}$ m$^2$ yr$^{-1}$. Our results suggests that the diffusion of sulfate ions within volcanic peaks

is relatively fast initially, perhaps through the inter-connected vein network, but slows significantly after 40 kyr. In the absence of clear evidence for a controlling environmental factor on sulfate diffusivity with depth/age, we hypothesize that the rapid decrease in diffusion rate from the time of deposition to ice of 50 ka age may be due to a switch in the mechanism of diffusion resulting from the changing location of sulfate ions within the ice microstructure.

## 1 Introduction

Records of chemical impurities within ice cores are frequently used as climate proxies (e.g., Wolff et al., 2010) or utilised in ice core dating via the identification of seasonal fluctuations (e.g., Sigl et al., 2016) or deposition from volcanic eruptions (Sigl et al., 2013). The assumption that chemical impurity signals have not undergone any significant post-depositional alteration is typically implicit in these applications. However, we know that post-depositional alteration of ice chemistry does take place and that this can become increasingly important as older, deeper ice is considered.





In this study we focus on one post-depositional process, the diffusion of chemical ions driven by concentration gradients within the ice structure. As we will explore, the strongest evidence for the diffusion of chemical signals in ice cores is seen in sulfate records. Volcanic sulfate peaks located in deep, old ice are typically low amplitude and span a wide age range relative to their recently-deposited counterparts (Fig. 1). Barnes et al. (2003) quantified the diffusion rates of sulfate and chloride ions in the Holocene ice of the Antarctic EPICA Dome C (EDC) core as $3.9 \times 10^{-8}$ $m^2$ $yr^{-1}$ and $2.0 \times 10^{-7}$ $m^2$ $yr^{-1}$ respectively. We note

that the incorrect value for sulfate ($4.7 \times 10^{-8}$ $m^2$ $yr^{-1}$) was quoted in the abstract of this study. Please see their Sect. 2.3 for their results. Two mechanisms enabling solute diffusion along veins or grain boundaries were proposed, both linked to the evolution of ice grain growth that promotes connections between veins. Note that here we consider ice grains to be surrounded by grain boundaries that intersect at triple junctions, which are populated by veins (see Fig. 2e of Ng (2021)). Soluble, ionic, impurities tend to be concentrated along grain boundaries and within veins (Barnes et al., 2003b; Bohleber et al., 2021; Mulvaney et al.,

1988), where liquids can exist at temperatures below zero. The eutectic temperature of sulfuric acid (the origin of the majority of sulfate within polar ice) is -73°C, which is well below the coldest ice sheet temperatures on Earth, meaning sulfate ions are likely to be mobile.

In a high profile study, Rempel et al. (2001) argued that 'anomalous diffusion' of chemical ions along temperature gradients can occur within veins. The implication was that a chemical signal within the ice sheet could be advected vertically, displaced

by as much as 50 cm depth relative to its location at deposition, while maintaining a similar amplitude. Such a process would have major consequences for cross-matching events, such as volcanic eruptions, between ice cores for stratigraphic purposes. Ng (2021) revisited this theory and challenged the impact of this phenomenon by demonstrating the importance of a neglected process: the Gibbs-Thomson effect. Ng argues that chemical ions present in the veins (grain boundaries and grain interiors are not considered) quickly diffuse so that peaks will be damped and broadened, and hence not survive into deep ice. For the ice

core community, Ng's revision means that chemical signals present in the veins will not be displaced in age/depth, but they will be destroyed over time.

In the context of international projects such as the Beyond EPICA Oldest Ice core (BE-OIC), which hopes to recover an Antarctic ice core dating back 1.5 Ma (Lilien et al., 2021), it is now critical to further quantify the rate at which chemical diffusion occurs in order to predict the extent of signal preservation at depth. Further constraints on the rates of chemical

diffusion could be crucial to understanding the mechanism(s) of chemical diffusion in polar ice. However, we highlight upfront that our current chemical measurements on ice provide bulk chemical concentrations and do not allow us to partition ions by location within the ice structure, i.e., grain boundary, vein or grain interior. New sub-millimetre scale measurement techniques will undoubtedly help in the future (Bohleber et al., 2021).

In this paper, we analyse volcanic sulfate signals in the EDC ice core (EPICA community members, 2004) and quantify time-

dependent effective diffusion rates. This work extends the time interval considered by Barnes et al. (2003) to well beyond the Holocene, incorporating four glacial-interglacial cycles. In order to quantify diffusion rates, we also simulate the impact of ice thinning resulting from ice flow on the preservation of chemical signals. Our work complements the recent study of Fudge et



al. (2022) (currently in pre-print, accepted) who used the same sulfate dataset but applied a different method to estimate effective diffusion rates. Our results are compared in Section 5.

## 2 Volcanic sulfate peaks in EDC ice core

Sulfate is deposited on the ice sheets as sulphuric acid by individual volcanic eruptions and can be distinguished in ice core sulfate profiles as sharp, distinct peaks lasting only a few years. Volcanic sulfate peaks are many times greater in magnitude than the sulfate background, which is dominated by marine sources. Furthermore, these easily identifiable signals occur regularly through time (Wolff et al., 2023), meaning evolution of their form resulting from diffusion and ice thinning can be traced down-core. In this study, we use EDC sulfate data measured using fast ion chromatography (FIC) (Severi et al., 2015; Fudge et al., 2023).

We quantify the rate of sulfate diffusion by comparing the shape of older volcanic peaks to their shape at deposition. This is possible because major volcanic sulfate peaks have a well-understood, reproducible peak shape very shortly after deposition, specifically a reproducible duration of deposition or peak width and a Gaussian form. Over the satellite era, two major volcanic eruptions have strongly perturbed the stratosphere, El Chichon in 1982 and Pinatubo in 1991 (Thomason et al., 2018). In both cases the loading of aerosol in the stratosphere increased rapidly to a maximum a few months after the eruption, and then decreased to background over a period of about 5 years, with an e-folding time of about a year (McCormick et al., 1995). Modelling studies suggest that this e-folding time varies between 6 months to 1 yr depending on the height, latitude and magnitude of the eruption (Marshall et al., 2019). Modelling of the deposition of material from the large 1815 CE Tambora eruption on the Antarctic ice sheet suggests that deposition should occur over 3-4 years, with full width at half maximum (FWHM) values of 1-2 yr (Marshall et al., 2018). Data from ice cores from regions of high snowfall, such as WAIS Divide in Antarctica, agree with these observations and model predictions. Sulfate signals associated with the largest eruptions of the Last Millennium last for about 3 yr (Koffman et al., 2013; Sigl et al., 2013). In the EDC ice core, volcanic peaks within the firn seem to be slightly wider (in terms of time) than those at WAIS Divide. The peaks for the 1257 CE eruption (unknown volcano, Fig. 1) and for Tambora (1815 CE, not shown) have FWHMs between 2 and 3 yr, with an observable increase in deposition covering about 5 yr. This slightly extended width of the peak at EDC might result from a combination of mixing of snow from different layers due to snowdrift, and surface roughness, which is significant compared to the low snow accumulation rate.

Barnes et al. (2003) already observed that volcanic sulfate peaks broaden (in terms of time) as they age and deepen through the Holocene in the EDC ice core. Here we extend our observations back to 450 ka. Figure 1 shows EDC volcanic sulfate peaks with a similar (within 30%) deposition flux to the value calculated for the 1257 CE eruption (83.9 mg m$^{-2}$), which implies that their initial shape at deposition would have been similar. While the examples in Fig. 1 show only a few snapshots, it is clear that the sulfate peaks continue to reduce in amplitude and to broaden (in terms of age range spanned) beyond the Holocene. One can compare the width of each peak with the horizontal bar on each plot representing 5 years of ice





95    accumulation. The FWHM, estimated as < 3 yr for the 1257 CE eruption, is around 10 years by 68 ka, and is almost 30 yr at

364 ka. These observations confirm that some form of chemical diffusion occurs at EDC, causing the sulfate peaks to be spread

over a greater equivalent time period as the core ages.

**Figure 1: Selected volcanic sulfate peaks in the EDC ice core showing changes in peak shape with depth. All peaks shown have a total sulfate flux of 63.7–91.0 mg m⁻² and in each case the x-axis spans a depth range of 1 m and the y-axis has the same scaling. The horizontal black bar on each plot indicates the depth range equivalent to 5 yr of ice accumulation at that depth in the core.**

We note that during glacial periods when the snow accumulation rate is low relative to interglacial periods, a 3 yr duration

sulfate peak would have covered a smaller depth range at deposition, and the sulfate peak height for the same magnitude

volcanic event would have been higher (assuming dry deposition dominates, the same amount of sulfate is deposited but across




a narrower depth range consisting of a smaller mass of snow). We also note that the 1257 CE eruption peak is in low density firn and therefore its ice equivalent depth coverage would be about 63% of the snow depth shown.

Figure 1 also illustrates the impact of ice thinning. Although the sulfate ions in older peaks have clearly diffused to dampen and broaden those peaks with respect to age, the width of the sulfate peaks in terms of depth does not alter much with depth/age

in the EDC core (all the x-axes on Fig. 1 span 1 m). Typically, the peaks cover a depth range of about 20 cm of ice, with a FWHM of about 10 cm. Ice thinning effectively compresses annual ice layers and narrows the depth range spanned by an individual chemical peak. However, it appears that at EDC, the combined action of peak broadening through chemical diffusion and peak narrowing via ice thinning results in a relatively constant peak width in the depth domain for the first 200 kyr. At other ice core sites, with different age-depth relationships and thinning functions, the evolution of peak shape with depth will

likely be different.

The evolution of volcanic sulfate peak shapes with depth therefore represents a convolution of the impacts of both the diffusion of sulfate signals along concentration gradients and the thinning of the ice sheet in which a volcanic sulfate signal is hosted (Table 1).

**Table 1: Summary of impacts of diffusion and thinning on peak shape.**

|  |  | **Peak height** | **Peak width** |
| --- | --- | --- | --- |
| **Diffusion** | **Age domain** | Decrease | Increase |
|  | **Depth domain** | Decrease | Increase |
| **Thinning** | **Age domain** | No change | No change |
|  | **Depth domain** | No change | Decrease |

## 3 Methodology

Our aim is to constrain the effective diffusion rate of sulfate over time in the EDC core by modelling the evolution of volcanic sulfate peak shapes with depth/age, accounting for the impacts of both chemical diffusion and ice thinning. To achieve this,

we first identified the volcanic peaks present in the EDC core, then generated equivalent unthinned, undiffused 'input' or 'deposited' peaks. These peaks were fed into a forward model, which simultaneously diffused them using a range of diffusion rates and thinned them according to a simple ice flow model. Comparison of each modelled (diffused and thinned) peak with the peak preserved in the ice core (that has also been diffused and thinned) allowed the optimum diffusion rate to be selected




for each peak. Figure 2a displays a flow chart of our methodology, which we describe in more detail in the following sections,
and Fig. 2b illustrates some of the key parameters involved.

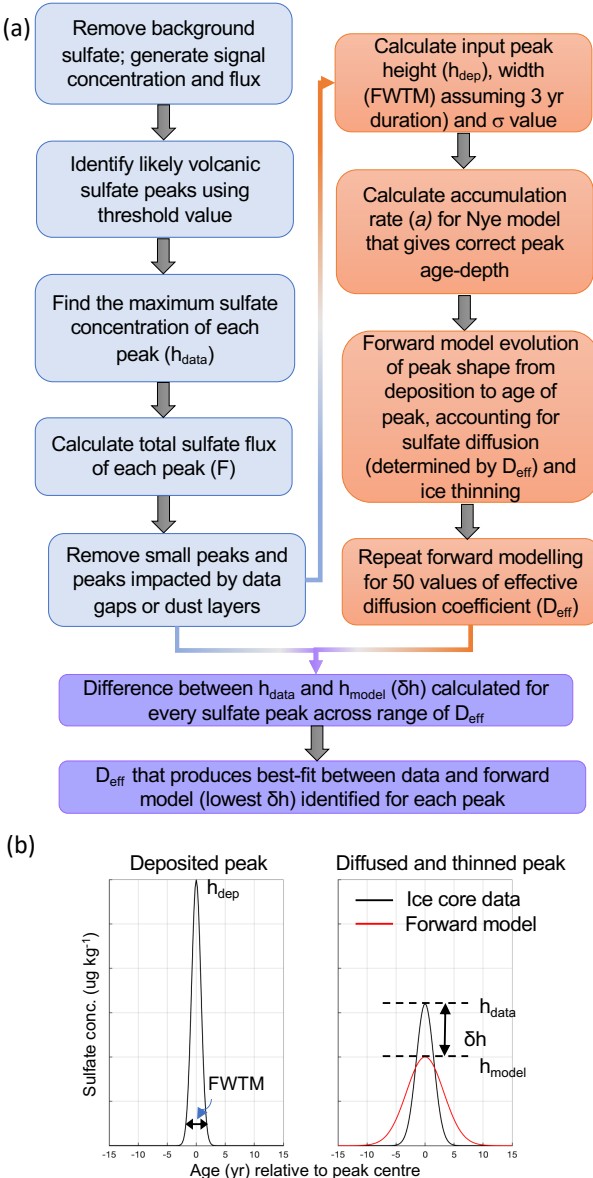

**Figure 2: Illustration of methodology. (a) Flow chart of methodology involving analysis of EDC sulfate data (blue boxes), generation of equivalent deposited peaks and forward modelling (orange boxes), and data-model comparison to find effective diffusion coefficient (purple boxes); (b) Example of a sulfate peak preserved in the ice core (right-hand side, black curve), its equivalent**
**deposited peak (left-hand side), and a peak produced by the forward model (right-hand side, red curve). In this example, the forward model produces a peak that is lower in amplitude and broader than the ice core data peak, suggesting that the effective diffusion coefficient used was too high.  Key parameters defined in the text are labelled.**



### 3.1 Identification of volcanic sulfate peaks

The EDC sulfate concentration data measured by FIC have a typical resolution of 5-6 cm in the top 100 m, 3-5 cm in the interval to 770 m, and 2 cm from there to the base of the core. The long-term background signal was removed from the sulfate data by subtracting a 200 yr moving median. Following Wolff et al. (2023), we multiply the residual sulfate concentrations between 0 and 358.6 m by (1/0.7) to account for a calibration discrepancy identified between FIC and standard ion chromatography measurements on EDC ice. Note that although this adjustment will impact our quantification of peak height

and peak area in this depth interval, there is no impact on the estimation of diffusion rate. To calculate the sulfate flux we multiplied the residual sulfate concentrations by the accumulation rate (Bazin et al., 2013).

To identify high amplitude volcanic peaks in the residual concentration data, we used a peak-finding algorithm (Matlab *findpeaks*) with a peak height threshold linked to the level of background variability in each 10 kyr time bin and a minimum time interval between adjacent peaks of 30 yr. For each volcanic sulfate peak identified and retained, we calculated the total

sulfate flux ($F$) by integrating the flux data with respect to time, between the peak edges set by local minima in the background data. All sulfate peaks with a total flux < 25 mg m$^{-2}$ were discarded (1027 out of 1618 events removed) to avoid using small peaks that may result from background signal variability and to limit the bias towards larger magnitude events with depth. This sulfate flux is just under half the magnitude of the Tambora 1817 CE eruption in the EDC record. Identification of enough volcanic peaks for analysis became difficult >450 ka.

Traversi et al. (2009) identified anomalous narrow, high amplitude sulfate peaks in the EDC record, speculated to result from the migration of sulfate into specific horizons within the ice. Each of these peaks are bordered by regions of relative sulfate depletion, hence the hypothesis that sulfate has moved, even been "sucked", into the peak horizon (see Fig. 3 of Wolff et al. (2023)). Although Traversi et al. (2009) found these peaks first appeared beyond 2800 m depth (~450 ka), we found some similar peaks during our analysis of shallower/younger ice. They caused a problem for our analysis because they were

identified as volcanic by our peak finding code but clearly do not result from diffusion. In order to objectively identify these anomalous peaks, we take advantage of the fact that they tend to be associated with sharp, high spikes in dust content. All peaks below 2100 m (204 ka) were checked for an anomalously high dust peak at the same depth. If a coincident dust peak was found, then that sulfate peak was removed from analysis (43 in total).

Lastly, we removed any peaks that were impacted by data gaps due to missing samples. Peaks between 0 and 770 m with a

data gap >14 cm and peaks between 770 m and 2800 m with a data gap > 6 cm within its FWHM depth interval were excluded (10 peaks). After all these filters were applied we had 537 peaks remaining.

### 3.2 Generation of input peak shapes

For a Gaussian function, the Full Width at Tenth Maximum (FWTM) is given by Eq. 1 and the area ($A$) under the peak is given by Eq. 2.



*FWTM = 4.292 * σ,* where $\sigma$ = standard deviation of Gaussian function.                    (*1*)

*A = h*σ/0.3989,* where *h* is the peak height maximum.                                           (*2*)

This leads to a relationship between FWTM, *h* and *A*:

*A=h*FWTM/1.712*                                                                                  (*3*)

We have measurements of sulfate concentration (*C*, $\mu$g kg$^{-1}$) with depth that have been converted to annual flux (*AF*, $\mu$g m$^{-2}$

yr$^{-1}$) using the accumulation rate (*a*, kg m$^{-2}$ yr$^{-1}$) via Eq. 4. We have also calculated the total sulfate flux (*F*) as the area (*A*)

under each peak in annual sulfate flux.

*AF = C * a*                                                                                      (4)

To obtain the initial peak, we assume that the total sulfate flux *F* of the peak has not changed since deposition and that the

FWTM of the initial deposited peak was 3 yr (see Sect. 2). Equation 3 then gives us the peak height maximum (*h*) of the initial

peak in units of annual flux ($\mu$g m$^{-2}$ yr$^{-1}$), which can be used in Eq. 4 to give us the peak height maximum of initial peak ($h_{dep}$)

in units of concentration that is an input to our forward model. The depth-equivalent in metres of 3 yr FWTM is calculated via

Eq. 5 in order to obtain $\sigma$ (see Eq. 1).

*FWTM (m) = 3 yr * a / ρ,* where $\rho$ is density in kg m$^{-3}$.                                  (5)

The duration of volcanic sulfate peak deposition may vary by a few years, potentially impacting our estimate of the effective

diffusion coefficient. For this reason, all peaks younger than 60 ka were additionally run through the forward model with input

peak widths equivalent to 1 yr and 5 yr. Preliminary testing showed that the choice of input peak width had a negligible impact

on the estimated effective diffusion coefficient for older peaks.

### 3.3 Forward model of chemical diffusion and ice thinning

We modelled the change in concentration of sulfate ions (*C*) with time (*t*) along the depth axis of sulfate peak (*x*) as ions

diffuse along a concentration gradient according to Fick's second law (Eq. 6), where $D_{eff}$ is the effective diffusion coefficient

in units of m$^2$yr$^{-1}$.

$$\frac{dC}{dt} = \frac{d}{dx}\left(D_{eff}(t).\frac{dC}{dx}\right)$$                                (6)

In Fickian diffusion, the medium through which a substance is diffusing has a uniform velocity profile, which is not the case

for the ice sheet at EDC. The ice dynamics at EDC are relatively simple though—the EDC core was drilled close to the ice

divide so the ice experiences very limited lateral flow (Legresy et al., 2000). By assuming no lateral flow, the dynamics of the

ice sheet consist only of one-dimensional (vertical) flow, with ice layers thinning with increasing depth and pressure. An

additional term is introduced to the diffusion equation to describe the change in sulfate concentration over time due to thinning,

which uses the vertical velocity (*v*) (Eq. 7).

$$\frac{dC}{dt} = \frac{d}{dx}\left(D_{eff}(t).\frac{dC}{dx}\right) - v(x,t).\frac{dC}{dx}$$       (7)





Vertical velocity ($v$) was estimated using a Nye model (Nye, 1963). This simple ice flow model predicts that layer thickness ($\lambda$) equals the accumulation rate ($a$) at the surface and is zero at the bed, and changes linearly in between, i.e., the thinning rate is constant with depth (Eq. 8, where $z$ is height above the bed and $H$ is the ice sheet thickness). The Nye model ignores density changes and assumes no melting at the bed.

$$\frac{\lambda}{a} = \frac{z}{H} \tag{8}$$

At steady state the vertical velocity will therefore equal the layer thickness at any depth (Eq. 9).

$$-v = \frac{\delta z}{\delta t} = \frac{za}{H} \tag{9}$$

Equation 10 then describes the change in vertical velocity with depth.

$$\frac{\delta v}{\delta z} = -\frac{a}{H} \tag{10}$$

In order to generate the velocity ($v$) required in Eq. 7, we assume that dv/dx is constant within the width of the peak, so that
the velocity at any distance ($x$) relative to that at the centre of the frame of reference is simply given by Eq 11.

$$v(x,t) = -\frac{ax}{H} \tag{11}$$

The Nye model was effectively tuned to the AICC2012 chronology (Bazin et al., 2013) by selecting a value of $a$, from within the range used by Parrenin et al. (2007), that produced the correct depth ($z$) of an ice layer of a given age ($t$) (Eq. 12). $H$ was kept constant at 3165 m, which is the mean ice sheet thickness over time interval considered here at EDC, according to Parrenin
et al. (2007). The more complex Lliboutry model used by Parrenin et al. (2007) is certainly a more precise description of thinning, but a tuned Nye model mimics it closely in the top two-thirds of the ice sheet (where thinning is close to linear), and its use is justified by the considerable reduction in computational complexity for this problem.

$$z = He^{-a/H} \tag{12}$$

For each sulfate peak, $h_{dep}$ and $\sigma$ of the initial deposited peak were used to generate a Gaussian distribution of sulfate
concentration along length $x$, which was then diffused and thinned via Eq. 7. Equation 7 is solved using a partial differential equation solver (PDEPE solver, MATLAB).

This forward model can be run at regular time intervals to produce a 3D representation of sulfate peak evolution through time and space (Fig. S1). It is the profile of sulfate concentration produced in the final time step that is equivalent to the diffused, thinned peak selected in the EDC dataset.

**3.4 Identification of effective diffusion rates**

For each volcanic peak identified in the data, the forward model was run using 50 effective diffusion rates log-spaced between $10^{-9}$ and $10^{-6}\,\mathrm{m^2yr^{-1}}$. As described above (Section 3.2) peaks <60 ka were run for 3 different values of input peak duration (1, 3 and 5 yr). For each individual simulation, the absolute difference between the modelled maximum peak height sulfate concentration ($h_{model}$) and the peak height of the selected sulfate concentration data peak ($h_{data}$) was calculated ($\delta h$, Fig. 2B)



and saved each time. For each volcanic event, we find the effective diffusion rate that produces the best-fit between modelled peak and data peak, the lowest value of δh.

## 4 Results

### 4.1 Effective diffusion rates of sulfate in EDC ice

Our forward modelling provides an estimate of effective diffusion rate, that is, the time-weighted rate of diffusion over the
entire history of the peak, for every volcanic event identified in the EDC sulfate record. To minimize the effect of any one peak, we have calculated the median effective diffusion rate across all the peaks in each 10 kyr or 20 kyr time bin (Fig. 3). The median absolute deviation (MAD) of the effective diffusion rate across all the peaks in each time bin is calculated to provide an uncertainty envelope (vertical bars on Fig. 3). Calculating the mean and standard deviation within each time bin produces a similar result (Fig. S2). Effective diffusion rates range from $2.4 \times 10^{-7}$ $m^2yr^{-1}$ for 0–10 ka to $1.6 \times 10^{-9}$ $m^2yr^{-1}$ for 410–430
ka.

Our results suggest there is a significant decrease in effective diffusion rate with age in the EDC ice core. Median rates are fastest in the Holocene ice ($2.4 \times 10^{-7}$ $m^2yr^{-1}$) and decrease sharply to $1.7 \times 10^{-8}$ $m^2yr^{-1}$ by 40–50 ka. Effective diffusion rates are significantly higher in the first 40 kyr of the EDC record relative to the remaining time. From 50 ka onwards, diffusion rates are an order of magnitude lower than in the Holocene, remaining consistently around $1.0 \pm 0.31 \times 10^{-8}$ $m^2$ $yr^{-1}$ until 200
ka.

Between 200 ka and 240 ka, the median effective diffusion rates appear to increase slightly, up to $1.8 \times 10^{-8}$ $m^2yr^{-1}$. This would imply that diffusion rates in ice of this age are higher than in younger ice, which would surely require a concurrent change in a controlling variable such as in temperature or chemistry. This possibility will be explored in Section 5.1. The overall trend from 240 ka to 450 ka is negative, implying that effective diffusion rates reduce with age/depth. However, the relatively high
spread of Deff values and subsequent high MAD values for several time bins, relative to the younger portion of the core (Fig. 5) make it difficult to have complete confidence in this feature.

Broadly speaking then, effective diffusive rates are relatively fast in the Holocene and into the Last Glacial Period but then appear to stabilise or possibly continue to decrease at a much-reduced rate back in time. If we consider again that these rates are the product of the evolution of diffusion rate over time, and assuming that all peaks are subjected to higher rates of diffusion
in the first 40 kyr of their history, the actual rate of diffusion occurring in older portions of the core must be considerably lower than displayed on Fig. 3. A stabilisation or plateau in effective diffusion rate does not mean that zero diffusion is occurring in ice of that age. It indicates that the diffusion rate has not changed since the last time window.

Looking only at our time-binned effective diffusion rates, there is no apparent systematic difference between ice deposited in a glacial period versus an interglacial period (Fig. 3, red shading indicates interglacials). Excluding the 0–45 ka time interval,
the median effective diffusion rate for all the sulfate peaks in glacial periods is $8.3 \pm 4.2 \times 10^{-9}$ $m^2$ $yr^{-1}$ (± MAD). The median



value is slightly higher for interglacial periods at $1.1 \pm 0.36 \times 10^{-8}$ m$^2$ yr$^{-1}$ ($\pm$ MAD) but the MAD envelopes overlap meaning there is no significant difference between the two.

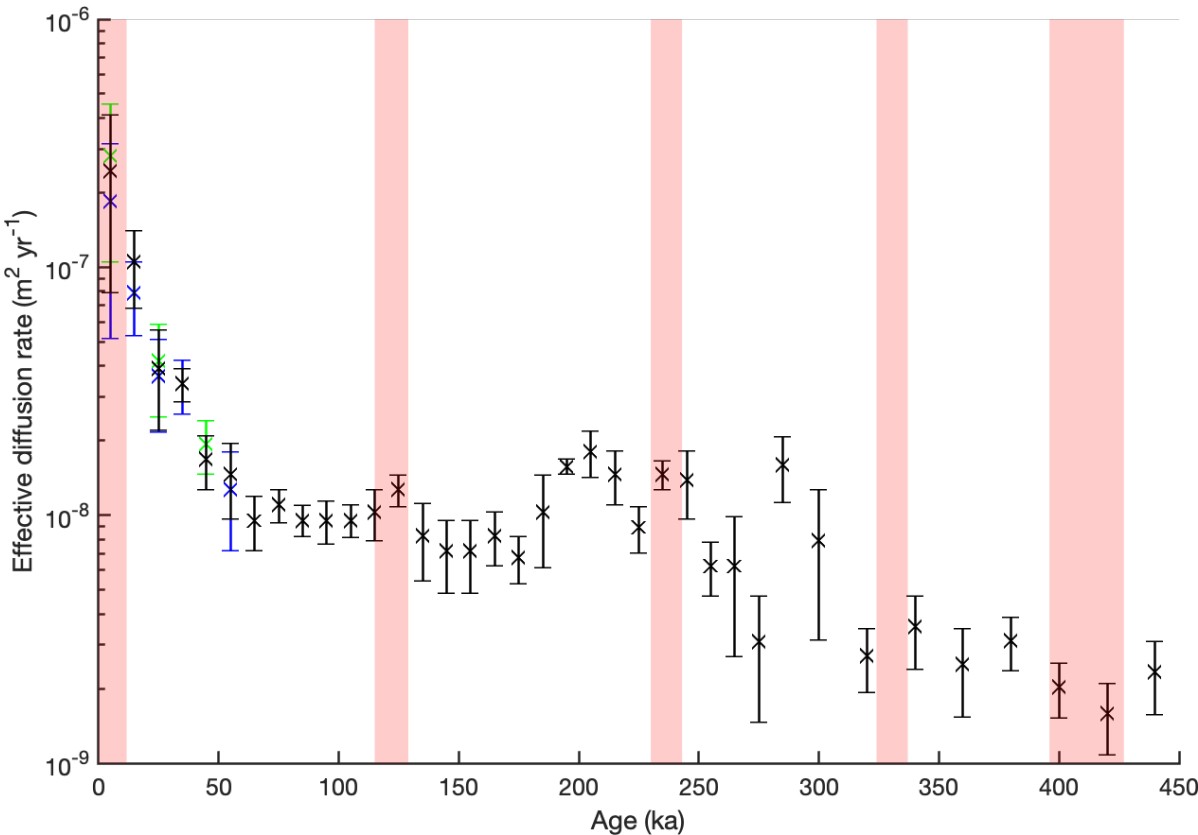

**Figure 3: Effective diffusion rates of sulfate in the EDC core. For each time bin, the median (black crosses) and median absolute deviation (MAD, black vertical bars) of effective diffusion rates across all volcanic events are shown. For the first 6 time bins, median and MAD of effective diffusion rates are also shown for volcanic peaks with FWTM of 1 yr (green) and 5 yr (blue). Varying peak width has a negligible impact on results from older time bins. Time bins are 10 kyr duration except those >300 ka, which are 20 kyr in duration. Light red shaded regions denote interglacial periods.**

## 4.2 Model validation

Before further interpretation of our results, we validate our forward modelling approach by testing if the model can simulate sulfate peaks similar to those in the EDC record using the effective diffusion rates calculated for the enclosing time bin (Fig. 4). The model performs well for the sulfate peaks shown at 68 ka, 294 ka, 394 ka and 437 ka. The range of peak heights simulated by the model (between the upper and lower bounds) include the height of the sulfate peak in the ice core. For the 137 ka peak, the model appears to slightly overestimate the rate of sulfate diffusion that has occurred. However, the mismatch is slight and if Fig. 4 is re-plotted using peak-specific effective diffusion rates calculated by the model (Fig. S3), the match between model and data is excellent for every peak, as one would expect. Several of the modelled sulfate peaks, e.g., 394 ka,





appear to be broader than the ice core sulfate peaks. As thinning does not impact peak width in the age domain (Table 1) this suggests that diffusion is under-estimated, or more likely, that the distribution of sulfate is not strictly Gaussian.

We further validate our modelling approach by running it with effective diffusion rates derived from the literature (Sect. 5.2) to observe the different peak shapes produced with a rate of $D_{eff}$ values, with (Fig. S4) and without (Fig. S5) the inclusion of thinning.

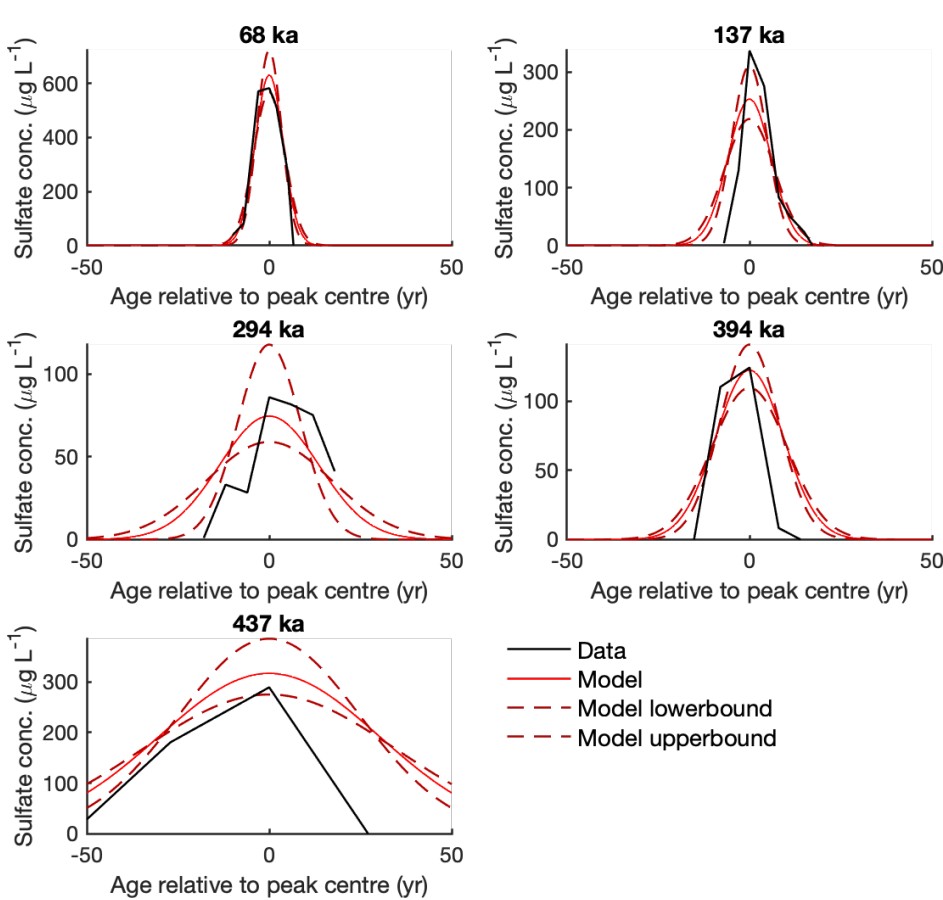

**Figure 4: Comparison of sulfate peaks in ice core with peaks produced by forward model. EDC sulfate peaks (black) of different age (indicated by bold titles) are compared to forward model simulations (red) produced using the median effective diffusion rate for 10 kyr time bin (or 20 kyr time bin if peak age >300 ka). Model upper- and lower-bounds (dashed dark red) are product of the median ± MAD effective diffusion rate (as Fig. 3).**

## 5 Discussion

This study suggests that the diffusion rate of sulfate at EDC is relatively rapid in the first 40 kyr and slows down to a quasi-constant value from then onwards. Here we will discuss the various factors that could contribute to this temporal trend, compare





our results to previous work and discuss the implications of this temporal variation in diffusion rate for potential mechanisms of sulfate diffusion.

### 5.1 Factors potentially influencing diffusion rate

At EDC sulfate ions can be dissolved in liquid within veins and at grain boundaries because the ice sheet temperature (Fig. 5d) is always above the eutectic temperature of -73°C of sulphuric acid (Gable et al., 1950). Although some sulfate may be present within the ice lattice at East Antarctic sites (Ohno et al., 2005), it cannot account for the majority of sulfate at EDC because, as Barnes et al. (2003) explained, self-diffusivity within ice is much slower than the effective diffusion rates we observe. The borehole temperature at EDC increases with depth from -52°C at the surface to -12.7°C at 2800 m, the deepest ice considered

here, due to the geothermal heat flux from below. As diffusion is a temperature dependent process, we might expect effective diffusion rates to increase with depth/age in the ice core but the opposite trend is seen in our data.

Increasing temperature with depth in the ice sheet impacts the ice microstructure in which sulfate ions are present because higher temperatures promote the growth of larger ice grains via normal grain growth (Durand et al., 2009). Barnes et al. (2003) proposed two mechanisms for sulfate diffusion that would both result in an increase in effective diffusion rate with increased

grain growth rate. However, a gradual transformation from <1 mm to >5 mm grains, with no marked change in overall grain growth rate, is observed through the section of EDC core considered here (Fig. 5b). Our analysis does not extend to the region with grain evolution is dominated by migration recrystallisation, which begins at ~3000 m when borehole temperatures rise above -10°C (Durand et al., 2009).

The process of normal grain growth is also known to be impacted by impurities, in particular insoluble particles, that effectively

pin the grain boundaries and limit the rate of grain growth (Durand et al., 2006). This grain boundary pinning is the reason for the sharp decreases in grain radius observed at EDC at each transition from low-dust interglacial ice to relatively high dust glacial ice (Fig. 5 b and c). Barnes et al. (2003) speculated the high dust concentrations of glacial periods would translate to reduced sulfate diffusivities due to the reversals in ice grain growth rate. In addition, Barnes et al. (2003) also speculated that sulfate ions would be less soluble and therefore less mobile in glacial periods relative to interglacials because sulfuric acid

would be neutralised through reaction with [carbonate-rich] dust particles. As stated in Sect. 4.1, we don't observe any significant difference between the effective diffusion rates calculated for glacial and interglacial ice, bar the Holocene. Again, we highlight the 50–200 ka interval, which spans the Last Interglacial as well as two glacial periods either side—the $D_{eff}$ values are consistently stable, around $1.0 \pm 0.31 \times 10^{-8}$ $m^2$ $yr^{-1}$.

However, something changes >200 ka when we observe more variable $D_{eff}$ values over time and within time bins. Looking at

the raw grain radius data used to calculate the mean value shown on Fig. 5b (which is all that could be found online) (Fig. 3 of Durand et al., 2009), the individual grain radii measurements are clearly more variable in this older age interval. Between 0-200 ka (~0-2000 m) the interglacial-glacial contrast in grain radius is ~< 0.5 mm, whereas in the 200-450 ka interval (~2000-2800 m) the raw data show repeated variations on the order of 4 mm. Even if the raw grain radius data could be compared to our $D_{eff}$ values, it is unlikely there would be a statistical relationship, but it seems fair to suggest that this marked increase in





the variability of grain growth rate with depth/age likely contributes to the increased range of $D_{eff}$ values predicted for ice >200 ka.

Finally, we examine whether or not the magnitude of the sulfate peak impacts any bias or trend in our results (Fig. 5a). There is no relationship between the total flux of sulfate and $D_{eff}$. Overall, it is difficult to attribute the variation in EDC sulfate effective diffusion rate observed with depth/age to any of the above factors.

Figure 5: Effective diffusion rate estimated for each EDC volcanic sulfate peaks compared to grain size and dust loading. (a) Effective diffusion coefficient ($D_{eff}$) estimated for each individual volcanic sulfate peak, colour-coded according to magnitude of total sulfate flux (see legend). Median $D_{eff}$ values and MAD range for each time bin are also plotted (blue crosses and vertical bars, as Fig. 3). (b) EDC grain radius (EPICA community members, 2004). (c) EDC dust flux (Lambert et al., 2008). (d) EDC borehole temperature profile (Buizert et al., 2021). Light red shaded regions denote interglacial periods.

## 5.2 Comparison to published values

Our model predicts a median effective diffusion rate of $2.4 \pm 1.7 \times 10^{-7}$ m²yr⁻¹ in Holocene ice (0–10 ka) with values for individual events ranging from $1 \times 10^{-6}$ to $5.2 \times 10^{-8}$ m²yr⁻¹. The previous estimate for effective diffusion rate of sulfate in



Holocene ice at EDC is $3.9 \pm 0.8$ x $10^{-8}$ m$^2$ yr$^{-1}$ (Barnes et al., 2003). This is significantly lower than both our median value for the 10 kyr interval and the rates implied by volcanic peaks around 10 ka only (Fig. 5): our median effective diffusion rate for 9–11 ka ice is $1.6 \pm 0.54$ x $10^{-7}$ m$^2$yr$^{-1}$ (n = 7). Fudge et al. (2016) estimated an even lower Holocene ice effective diffusivity of $2.2$ x $10^{-8}$ m$^2$ yr$^{-1}$ in the WAIS Divide ice core, which is a warmer location.

Beyond the Holocene, our estimate of $1.0 \pm 0.31$ x $10^{-8}$ m$^2$ yr$^{-1}$ for 50–200 ka is higher than the new EDC estimate for 0–450 ka ice of $5 \pm 2$ x $10^{-9}$ m$^2$ yr$^{-1}$ proposed by Fudge et al. (2022) and our uncertainty ranges don't overlap. However, if our more variable estimates for effective diffusivity in >200 ka ice are included, then our median effective diffusivity for the entire 50–450 ka interval is $8.3 \pm 4.2$ x $10^{-9}$ m$^2$ yr$^{-1}$, within range of Fudge et al.'s value.

Overall, these comparisons are encouraging, both for having confidence in our methodology and for confirming that previous estimates for sulfate diffusivity in EDC ice are not incredibly over- or under-estimated. Still, significant differences exist in our results, which given that the three EDC studies utilise the same sulfate data set, must originate from methodological differences.

Barnes et al. (2003) did not only target volcanic sulfate peaks in the EDC record but performed a windowed-differencing operation on the entire Holocene sulfate time series to quantify signal damping and fit a diffusion model to the resulting trend. The higher Holocene ice $D_{eff}$ produced by our method could therefore result from faster sulfate diffusion along the steep concentration gradients of volcanic peaks relative to the muted variations of background marine sulfate. This might actually be related to differences in location within the ice microstructure: in ice of lower concentration, the sulfate may mainly be accommodated in two-grain boundaries, which may be discontinuously connected. In volcanic peaks with much higher amounts of sulfate, at least initially the grain boundary or vein connectedness is more likely to be complete, leading to faster diffusion. An additional consideration is that Barnes et al. did not include the combined influence of diffusion and thinning. The sulfate record was 'unthinned' for the purpose of their calculations of sulfate gradients. For the majority of the Holocene there is little thinning but by 336 m (the deepest considered by Barnes et al.) a layer will be thinned to 93% of its original width (Bazin et al., 2013), meaning the sulfate concentration gradient will be steepened slightly, with the potential for faster diffusion.

Fudge et al. (2022) used two methods to calculate the change in sulfate variability down-core. First, they applied the same method as Barnes et al. (2003), which they refer to as 'scaled mean gradient', again on all the sulfate data. Second, they identify volcanic peaks to obtain the change in peak width over depth/time, then fit a 1D diffusion model (Fudge et al., 2016). This model differs from ours because it uses a fixed input/deposition width (in metres) for all peaks identified in interglacial and glacial periods, based on Holocene and LGM values respectively. It also does not solve for diffusion and thinning simultaneously.

**5.3 Implications of results**

One possible explanation for the high initial diffusion rates, is that the sulfate present in veins undergoes "Gibbs–Thomson" diffusion, modelled by Ng (2021), when first deposited. Ng predicts diffusion rates for ions in the veins that are rapid (2.1 x



$10^{-6}$ m$^2$ yr$^{-1}$), an order of magnitude faster than what we observe in Holocene ice. If the high concentrations of sulfate deposited from volcanic events are predominantly located within well-connected veins (as speculated above) then Gibbs-Thomson diffusion would provide a mechanism for the rapid initial diffusion. But then why does the diffusion rate decrease with

depth/age? Maybe Gibbs-Thomson diffusion is so efficient the sulfate concentration gradient within the veins is quickly reduced, leaving only diffusion along two-grain boundaries. Or maybe the veins that are initially well-connected and conducive to diffusion become less so over time. There is no obvious bias of larger sulfate flux events towards higher diffusion rates within Holocene ice or elsewhere, which would imply the diffusivity is not limited by the magnitude of the sulfate concentration gradient but more likely by the degree of connectivity within the ice microstructure.

As mentioned above, Barnes et al., (2003) presented two different mechanisms for sulfate movement in ice, a 'connected' vein or grain boundary model and a 'disconnected' vein or grain boundary model. Both predicted diffusion rates on a similar order to the rates we estimate in the older (>50 ka) ice: 5-7 x $10^{-8}$ m$^2$ yr$^{-1}$ for the connected model and 3 x $10^{-8}$ m$^2$ yr$^{-1}$ for the disconnected model. Without knowing more about the ice microstructure EDC, and the location of sulfate ions within it, it is difficult to favour one model over another. But both offer an alternative to the rapid Gibbs-Thomson diffusion and both can

(according to Barnes et al. 2003) operate in the veins or along grain boundaries.

We hypothesize that several different mechanisms of sulfate diffusion may operate across the depth/age range at EDC and that the predominance of one process over another is dependent on partitioning of sulfate within the ice microstructure. As such, it is the balance of rapid Gibbs-Thomson diffusion in the veins when sulfate is first deposited versus slower Barnes-types diffusion options at grain boundaries that dictates the evolution of effective diffusion rate with depth/age of the ice, rather than

any environmental factor such as the rate of grain growth or temperature.

**5 Summary**

Our results suggest that if a sulfate signal deposited at EDC survives the relatively fast initial diffusion (Holocene ice median $D_{eff}$ = 2.4 ± 1.7 x $10^{-7}$ m$^2$yr$^{-1}$) it will experience a much-reduced diffusion rate on the order of 1 x $10^{-8}$ m$^2$ yr$^{-1}$ or less from then on, at least until 450 ka. The high variability in our estimates >200 ka makes it difficult to determine if the diffusion rate stays

constant or continues to decline with age from 200 ka. In the absence of clear evidence for a controlling factor on sulfate diffusivity with depth/age, we hypothesize that the rapid decrease in diffusion rate from the time of deposition to ice of 50 ka age may be due to a switch in the dominant mechanism of diffusion resulting from the changing location of sulfate ions within the ice microstructure.

Our findings need to be confirmed by analysis of high resolution sulfate dataset from other ice cores.  It will be interesting to

compare the effective diffusivity profile of EDC with profiles from other cores that have similar or contrasting temperature profiles, chemical variations and ice microstructure variations. Finally, sulfate is just one chemical ion of interest in deep ice. There is an urgent need to constrain effective diffusion rates for different chemical ions.



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

volcanic sulfate peaks identified in this study and their modelled effective diffusion rates is provided in the Supplement.

**Author contributions** RHR and EWW designed the study. YQB performed initial data analysis with assistance on coding from PRFB. RHR wrote the paper and performed data analysis with input from YQB and EWW. MS provided EPICA Dome C sulfate data. All authors provided feedback on the manuscript.

**Competing interests** One author (EWW) is an Editor at Climate of the Past. The authors declare no other competing interests.