# Peer review of "New estimates of sulfate diffusion rates in the EPICA Dome C ice core"

_EGUsphere, 2024_

## Referee Comment (RC1)

**New estimates of sulfate diffusion rates in the EPICA Dome C ice core**

by Rachael Rhodes, Yvan Bollet-Quivogne, Piers Barnes, Mirko Severi, Eric Wolff

The manuscript reports on a detailed examination of volcanic sulfate peaks through a 450 kyr record from the EPICA Dome C ice core. Such compositional inputs are characterized as relatively high amplitude, short duration events; yet in older, deeper ice the amplitudes of recovered signals are observed to decrease and their apparent durations are extended by post-depositional processes. The authors' analysis points to a median effective diffusivity in Holocene ice that is somewhat higher than that found in early work by Barnes et al. (2003), but the median diffusivity in older ice is considerably lower and interpreted by the authors to possibly be caused by a switch in diffusion mechanisms, as might be expected from a change in impurity locations within the polycrystalline microstructure. This careful and interesting study will be of considerable interest for its importance in cataloging the post-depositional alteration of soluble impurity anomalies, thereby influencing future interpretations of soluble impurity records in older ice cores. The scientific approach and applied methods appear to be well-considered and the results and conclusions are presented in a balanced and clear manner. Following minor revisions, I anticipate that the manuscript will be ready for publication. I offer several comments below for the authors' consideration, with a view to clarifying some of their reasoning and encouraging firmer connections to the physical processes involved.

The discussion of previous theoretical results in the introduction is appropriately concise, but might be altered slightly to some benefit. On line 45, the 50 cm displacement noted in the abstract of the paper by Rempel et al. (2001) is quoted without sufficient context to enable the reader to understand the conditions that led to that particular figure, which applied to ice of Eemian age in the GRIP ice core. Since that theory would predict different displacements in the EPICA Dome C core at different depths, I'd suggest rewording to something like: "The implication was that a chemical signal of Eemian age in the GRIP ice core ...". Regarding the following sentence, a half meter doesn't seem like much in a >2 km deep core, so it isn't clear whether the consequences for cross-matching events between ice cores for stratigraphic purposes would in fact be "major", or typically quite minor – perhaps the adjective should be removed. On line 50, the claim is made that Ng's (2021) modified theory would prevent such compositional displacement, but destroy the chemical signals over time. This would be somewhat unsatisfying given that chemical

signals clearly persist for long durations, but Ng does show that deep signals can remain intact if the effective diffusivity is reduced or if spatial variations in grain size are invoked. I'd suggest appending the sentence with something like: "... they will be destroyed over time if they are free to diffuse unimpeded through connected veins into adjacent low concentration regions."

The differences between the Rempel and Ng treatments are not central to this manuscript, but as the former model is disregarded following the opening sentences of this paragraph, for further context I think it worth clarifying my own understanding of the primary difference between these formulations. Rempel et al. (2001) did not, in fact, ignore the Gibbs–Thomson effect, but instead made a particular (possibly naive) assumption regarding the relationship between impurity loading and grain size, or more precisely vein density (length of veins per unit volume). Citing the laboratory work of Mader (1992) and the theoretical treatment of Nye (1989), they reasoned that surface energy (i.e. the Gibbs–Thomson effect) would cause vein radii to evolve quickly towards a uniform value and that reported anti-correlations between grain size and impurity loading at GRIP are consistent with expectations if vein radii vary only over length scales that are much greater than those over which the bulk concentration of dissolved solutes vary. Notably, they did not explicitly treat the evolution of vein density (or crystal size) in their model, but application of their key uniform-vein-radius assumption would imply that the slow temporal variations in liquid fraction associated with the gradual displacement of bulk impurity signals that they predict requires vein density (and likely crystal size) to evolve at commensurate rates. In contrast, Ng (2021) argued that a physical mechanism for producing variations in vein density with impurity loading of the particular type assumed by Rempel et al. (2001) is both lacking and unlikely (for example, the cited study by Durand et al, 2006 argues against a causal relationship between grain size and soluble impurity content; I'm not aware of published empirical efforts to systematically quantify vein density as a function of bulk concentration at the scale of volcanic anomalies). In his preliminary model development, Ng (2021) assumes that vein density is effectively uniform or varies only much more gradually with depth than do compositional signals (i.e. the opposite assumption to that made by Rempel and coauthors). With bulk composition and vein density controlled by separate mechanisms in Ng's model (e.g. in illustrative calculations vein density is almost spatially uniform, with only extremely gradual changes promoted by grain growth over time), bulk concentration gradients lead to signal diffusion and destruction over time, as you note. This happens because the larger vein radii that are present where bulk concentration is higher gradually diminish in size as solute diffuses and enlarges vein radii in adjacent ice – evolving towards the uniform value for vein radius that Rempel et al. (2001) had assumed to be maintained instead through vein density (e.g. crystal size) changes. Ng also demonstrates the effects of alternative patterns of imposed crystal size (i.e. vein density) variation that can produce spurious peaks in bulk concentration. Importantly, since no feedback between vein density and bulk concentration changes is contained in Ng's model, impurity anomalies are effectively constrained to remain in high vein-density regions without a tendency

for the post-depositional translation that the Rempel et al. (2001) model predicts. To summarize: if vein density were to evolve in such a way as to keep vein radius constant, Ng's model would predict the same result as Rempel et al. (2001) – translation with negligible signal diffusion. Absent such fortuitous changes, Ng's model predicts ongoing signal diminishment and their evident resilience in ancient ice requires some other mechanism for retarding such changes – like disconnected veins or significant impurity loading within crystal interiors.

This has been a rather verbose digression. Perhaps rather than saying that the Gibbs–Thomson effect was neglected, it would be more correct to say something along the lines of: "... challenged the impact of this phenomenon by noting that since soluble impurity content appears not to exert a dominant control on ice grain size (e.g. Durand et al., 2006) and by extension, vein density, the Gibbs-Thomson effect should cause vein radii to adjust by producing solute concentration gradients that diminish bulk concentration anomalies."

The approximately Gaussian form of observed volcanic sulfate anomalies is somewhat curious (line 74). One might have expected fallout and deposition to be concentrated at first and subsequently diminish over time and so be "front-loaded" to some extent. Based on modern observations, could you comment on whether the Gaussian shape results from short-term post-depositional changes (e.g. due to Gibbs–Thomson diffusion), or whether this is instead the characteristic pattern of volcanic fallout from stratospheric levels?

In Figure 1, comparisons of the displayed scale bars showing $5\,\mathrm{yr}$ of ice accumulation with the observed sulfate peaks provide vivid illustrations of post-depositional effects. However, the $5\,\mathrm{yr}$ span collapses onto a vertical line in the final 3 examples. I appreciate that the text gives further context, with the quoted $30\,\mathrm{yr}$ span for the $364\,\mathrm{ka}$ peak. However, I'd suggest modifying the figure caption or annotating each panel with the number of years that the $1\,\mathrm{m}$ depth range represents.

I found the theoretical development in sections 3.2 and 3.3 somewhat confusing. The standard convention in the modeling literature with which I am most familiar is to treat equations as portions of sentences, with appropriate punctuation (e.g. see Ng's 2021 paper). Instead, here you refer to the equations by number, and subsequently separate them out from the text. To me, this seems disjointed. For example, I would favor a modification of the beginning of 3.2 to something like: "For a Gaussian function with standard deviation $\sigma$, the Full Width at Tenth Maximum is given by

$$FWTM \approx 4.292 \times \sigma \;, \tag{1}$$

while the area under a peak of height $h$ is

$$A \approx (h \times \sigma)\,/0.3989 \;, \tag{2}$$

so that

$$A \approx h \times FWTM/1.712 .\tag{3}$$

In equations (6) and (7) you note that the effective diffusivity is expected to be a function of time. However, my understanding is that your model calculations in fact treat the diffusivity as constant through time – is this correct? It wasn't immediately obvious to me how equation (12) came about and why there is no explicit dependence on time. Indeed, $a$ is really a rate, so I believe $at/H$ is needed in the argument of the exponential to ensure dimensional correctness, and integration of (9) would produces this result following correction of a sign error.

On line 248 temperature and chemistry are mentioned as controlling variables. Perhaps grain size, or more generally, microstructure, should be mentioned as well.

On line 295 the very low eutectic temperature of sulfuric acid is used to justify the expectation that sulfate ions can be dissolved in liquid at EDC temperatures. However, in the paragraph beginning on line 155 you mention the Traversi et al. (2009) finding that appears to suggest that sulfate reacts with dust to presumably form a solid precipitate. Would it possibly be worth saying more here about the potential effects of chemical reactions between different impurity species?

The brief discussion in 5.3 begins by noting that the simplest version of Ng's (2021) model predicts much faster diffusion than is observed in the Holocene ice, which itself is faster than that observed in deeper regions. That the inferred diffusivity does not appear to depend on signal size would also seem to differ from Ng's (2021) model predictions. The proffered suggestion that Gibbs–Thomson diffusion efficiently reduces vein concentration gradients would appear to effectively transform Ng's model to the Rempel et al. (2001) model, albeit only if vein density can evolve to enable signal translation. As the Barnes et al (2003) treatment relies upon effects of grain-size evolution, it perhaps might contain some of the essential elements that these other two models lack. I'm not sure I follow the reasoning behind the final sentence of this section. You've shown that the effective diffusion rate in the Holocene and early Pleistocene is both much slower than Ng's Gibbs–Thomson mechanism would predict and not systematically dependent on anomaly magnitude, so what makes you conclude that Ng's model correctly describes the controlling mechanism? I thought that I understood the Barnes-type model to depend on grain growth, but in the final clause you say that the rate of grain growth isn't important. Please clarify.

There's a typo in the title of the penultimate reference.

---

## Author Comment (AC2)

Reply to Reviewer 2, Jeffrey L. Kavanaugh

We are grateful to Jeffrey L. Kavanaugh for his detailed and thoughtful review of our manuscript. We are confident that all concerns raised can be addressed in a revised manuscript. In particular, the description of our modelling approach will be much improved thanks to his queries and suggestions.

We do not reproduce the entire review here – sections of it are shown in italics. We note that some equations are not well-reproduced here. Our replies are given in regular font. Reviewer 2 also uploaded an annotated version of the manuscript, containing editorial suggestions that we are grateful for and will largely adopt in a revised version.

1. Gaussian form

*…Their use here is more than justified*
*– but they remain just an approximation of the measured peak forms. (I'll also note that the description of stratospheric concentrations of sulphates following a major eruption (Lines 75–77) is decidedly non-Gaussian, being strongly asymmetric around the peak, and therefore it's reasonable for the reader to question whether sulphate concentrations in snow and ice are similarly asymmetric. Some additional discussion would help clarify this.*

A similar point was made by Reviewer 1 and our reply is repeated below. We note that the manuscript text states peaks are Gaussian in form "very shortly after deposition". We do not argue that the direct deposition of sulfate from the atmosphere is perfectly symmetric.

The reviewer is of course correct that the accumulation of sulfur in the stratosphere and its subsequent deposition to the ice sheet will be asymmetric, with a relatively sharp onset and a decaying tail. This asymmetry is indeed seen in recent eruptions, but observations of eruption signals at EDC shows that the signals tend towards symmetry within a time (of order 1 kyr) that is relatively short compared to the 400 kyr of this study. For this reason we chose the mathematical simplicity of assuming an Gaussian shape from the start. Our sensitivity studies (Fig 3) where we changed the width of the initial peak between 1 and 5 years show that the exact width or shape of the initial peak does not materially effect the derived effective diffusion coefficients.

*…it's unclear to me why FWHM is used to describe peak widths observed in the EPICA Dome C core throughout Section 2 (which describes the data), but FWTM is used throughout Section 3 (which describes the model). This isn't a major issue, to be sure (as the two are always related as FWTM/FWHM = 1.83), but it seems an unnecessary switch to make given that one value should be as easy to determine as the other from the data (but again, the data are presented in terms of FWHM, not FWTM). If FWTM is preferred, please include a brief explanation as to why.*

We agree, the use of FWHM and FWTM is confusing and not necessary. Section 2 can be modified to use FWTM, or more generally 'width'. FWTM is preferred for the model because it is closer to the 'width' metric of volcanic events in ice core sulfate identified by previous studies, e.g., Sigl et al., 2013, and therefore useful for comparison.

*Given that the objective here is to set up the numerical model, I recommend rearranging Section 3.2 somewhat, moving from expressing quantities in the time domain (FWTM/FWHM expressed in terms of years) to the spatial domain (FWTM/FWHM expressed in terms of distance) as quickly as possible, which could be accomplished by stating immediately stating that if the peak width is 3 years; moving Eq. 5 up to where it would become eq. 2; and then discussing relevant areas and fluxes (currently Eq. 2-4). (In my reading of this subsection, Eq. 2-4 needn't come before Eq. 5, but I might be missing something.)*

Section 3.2 will be re-written following this advice and advice from Reviewer 1.

2. Description of the forward model

Reviewer 2 suggests re-writing equations 6 and 7. We agree we should more correctly use partial differentials in Eq. 6 and 7. The reviewer asks us to include a more fundamental intermediate equation. We are not convinced this is necessary and consider it less confusing for the reader to present the equation we actually solve including the parameter we try to derive.

*Some clarification of the description of ice deformation at flow divides is also necessary. Lines 195-196 state "By assuming no lateral flow, the dynamics of the ice sheet consist only of one-dimension (vertical) flow, with ice layers thinning with increasing depth and pressure."*
*This misstates flow conditions in a couple of ways…*

We thank Reviewer 2 for his suggestions on how to improve the section on ice deformation. We will update a revised manuscript with more appropriate terminology and remove the inaccurate statements.

*It's important to also note that it is only because the same $\partial C/\partial x = 0$ and $\partial C/\partial y = 0$ conditions are met that diffusion (Eq. 6) can be treated as a 1-D problem here, rather than a 3-D one. With respect to Eq. 6, the effective diffusion rate $D_{((}$ is expressed as a function of time (i.e., $D_{(((}(t)$. However, given the description of the model in the text, it seems that the diQusion rate $D_{((}$ is held constant for each model run (with the best-fit diffusion rate for each sulfate peak determined independently from a set of 50 runs with log-spaced $D_{((}$ values). Is this correct?*

Yes, correct.

*If so, this would mean that the sulfate diffusion rate is determined by the time at which the snow fell, rather than by the length of time the snow is resident within the ice sheet – which has implications for the interpretations regarding Gibbs-Thompson diffusion vs. slower processes discussed in Section 5.3.*

The 'time at which the snow fell' and the 'length of time the snow is resident in the ice sheet' are in effect the same value because the units of ice age ('time at which the snow fell') is thousands of years before present, if we understand the Reviewer correctly here.

We agree that using a constant effective diffusion coefficient negates the possibility of testing how a time-evolving diffusion coefficient (perhaps due to changes diffusion processes) impacts the outcome. This will be clarified in a revised manuscript, also following Reviewer 1's comment. In a future treatment we could attempt to derive the time varying diffusion coefficient under certain assumptions by starting diffusion from various depths or by adopting a finite difference calculation method, but that is beyond the scope of this paper.

*Related to the discussion of the material derivative, it is not specified in the text whether the model is constructed in a Eulerian (i.e., fixed) coordinate system or a Lagrangian coordinate system (in which the coordinates track the deforming material). The framing of the equations suggests that a Eulerian coordinate system is used; this should be stated. (This is a relevant question because the sulfate peaks are advected downward through time.)*

A Eulerian coordinate system is used to generate a frame of reference relative to the centre of each peak.

*There are a few other concerns I have regarding the equations and phrasing in Section 3.3:*
*• The depth variable $z$ is defined (on Line 202) as the "height above the bed," but is subsequently referred to as "depth" (e.g. Line 205, which describes Eq. 9, and Line 207, which sets up Eq. 10). This is unnecessarily confusing to the reader, as thinking of $z$ as a depth-below-surface reverses the sign convention. It would be much clearer to refer to $z$ as "height above bed" throughout the text.*

Good point – manuscript will be modified.

*In Eq. 6-12, the spatial variable switches back and forth between $x$ and $z$. I suspect that $z$ refers to "depth within the ice sheet" and $x$ to "distance along the ice core," but didn't see this clarified in the text.*

Again, good point. Thanks. x is the distance along core relative to the centre of frame of reference (L210). Manuscript will be modified to clarify this.

*Eq. 8 is dimensionally incorrect: the left-hand side has units of $m/m\ yr)* = yr$, whereas the right-hand side is dimensionless (units: $m/m$); the equation is therefore a mathematical impossibility.*

We should have clarified that both the layer thickness lambda and the accumulation rate a are in m yr^-1.

*Equation 12 has a similar issue with dimensionality: the argument of an exponent must be dimensionless, whereas $-a/H$ has units of $yr)*$.*

Yes, the equation should indeed have at/H in the exponential – thanks for spotting this typo.

*The equation defining the downward velocity field (Eq. 9) has issues with sign Convention*

The equation suggested by the Reviewer will be included in a revised manuscript so that velocity (v) is positive.

*Eq. 9 defines the ice velocity as being downward (i.e., negative). The negative sign ahead of the velocity term in Eq. 7 (as written in the manuscript) would therefore result in an upward (i.e. positive) velocity field. This is why the second term on the right-hand side of the material derivative equations must be positive in both 3-D and 1-D forms of the expressions for the material derivative.*

We will correct the sign error to ensure consistency between Eq. 9 and 9-11.

**3. Significant figures**

We thank the Reviewer for his advice on significant figures, which we will follow in a revised manuscript.

**4. Section 5.3 Implications**

*I'm not convinced this material needs to be presented separately from that presented in Section 5.1 ("Factors potentially influencing diQusion rate"), as the discussion as to whether and when the slower Barnes [2003]-type diQusion or the more rapid Gibbs-Thompson Ng [2023]-type diQusion might operate seems to fit well within that general topic.*

We respectfully disagree with the Reviewer's opinion here. We will modify the sub-heading of Section 5.3 to make the distinction between Section 5.1 and Section 5.3 clearer.

*I'm also not sure whether the study directly addresses whether Gibbs-Thompson diQusion might explain initial (high) sulfate diQusion rates, but not later (lower) rates of diQusion. This relates back to my earlier question regarding whether the eQective sulfate diQusion rate $D_{((}$ is held constant for a given sulfate peak during the model runs. If $D_{((}$ is held constant throughout each model run, the model does not directly answer the question: no "old" ice would have been modeled with high initial (Gibbs-Thompson) diffusion, followed by lower (Barnes-type) diffusion.*

The Reviewer is correct in stating that the diffusion rate does not change as a function of time for each individual peak. Each peak is modelled with a constant effective diffusion rate (Deff). However, Deff does not represent the instantaneous diffusion rate in ice of that peak's age but a time-weighted rate of diffusion over the entire history of the peak (see L 234). By analysing each individual volcanic peak, and assigning the 'best-fit' Deff for each one, we are able to ascertain that effective diffusion rate (and therefore diffusion rate) decreases with time. We are not able to quantify the change in instantaneous diffusion rate with time.

In a future treatment we could attempt to derive the time varying diffusion coefficient under certain assumptions by starting diffusion from various depths or by adopting a finite difference calculation method, but that is beyond the scope of this paper. But the reviewer is correct, at present we have not modelled any ice as having a high initial rate followed by a lower rate at some depth, although this is what we suspect is happening.

---

## Author Response (AR1)

Line numbers in this document refer to the tracked changes version of our revised manuscript.

Reply to Reviewer 1, Alan Rempel.

We are grateful to Alan Rempel for his constructive and thoughtful review of our manuscript. We are confident that all concerns raised have been addressed in the revised manuscript. The Discussion section has improved thanks to his insights on the physical processes involved.

We do not reproduce the entire review here – sections of it are shown in italics. Our replies are given in regular font.

*The discussion of previous theoretical results in the introduction is appropriately concise, but might be altered slightly to some benefit. On line 45, the 50 cm displacement noted in the abstract of the paper by Rempel et al. (2001) is quoted without sufficient context to enable the reader to understand the conditions that led to that particular figure, which applied to ice of Eemian age in the GRIP ice core. Since that theory would predict different displacements in the EPICA Dome C core at different depths, I'd suggest rewording to something like: \The implication was that a chemical signal of Eemian age in the GRIP ice core ...".*

This suggestion has been adopted in revised manuscript (L60).

*Regarding the following sentence, a half meter doesn't seem like much in a >2km deep core, so it isn't clear whether the consequences for cross-matching events between ice cores for stratigraphic purposes would in fact be \major", or typically quite minor { perhaps the adjective should be removed.*

50 cm is indeed a small interval of depth but at the base of an ice core, where layers are thinned, it equates to a substantial amount of time. The word "major" has been removed as suggested.

*On line 50, the claim is made that Ng's (2021) modified theory would prevent such compositional displacement, but destroy the chemical signals over time. This would be somewhat unsatisfying given that chemical signals clearly persist for long durations, but Ng does show that deep signals can remain intact if the effective dffusivity is reduced or if spatial variations in grain size are invoked. I'd suggest appending the sentence with something like: \... they will be destroyed over time if they are free to dffuse unimpeded through connected veins into adjacent low concentration regions."*

This suggestion has been adopted in revised manuscript (L69-70).

*The differences between the Rempel and Ng treatments are not central to this manuscript, but as the former model is disregarded following the opening sentences of this paragraph, for further context I think it worth clarifying my own understanding of the primary difference between these formulations…This has been a rather verbose digression. Perhaps rather than saying that the Gibbs{Thomson ffect was neglected, it would be more correct to say something along the lines of: \... challenged the impact of this phenomenon by noting that since soluble impurity content appears not to exert a dominant control on ice grain size (e.g. Durand et al., 2006) and by extension, vein density, the Gibbs-Thomson e ect should cause vein radii to adjust by producing solute concentration gradients that diminish bulk concentration anomalies."*

We thank the Reviewer for taking the time to detail his insightful thoughts. We will considered them carefully and modified the manuscript in response (L63-66).

*The approximately Gaussian form of observed volcanic sulfate anomalies is somewhat curious (line 74). One might have expected fallout and deposition to be concentrated at rst and subsequently diminish over time and so be \front-loaded" to some extent. Based on modern observations, could you comment on whether the Gaussian shape results from short-term post-depositional changes (e.g. due to Gibbs{Thomson dffusion), or whether this is instead the characteristic pattern of volcanic fallout from stratospheric levels?*

The reviewer is of course correct that the accumulation of sulfur in the stratosphere and its subsequent deposition to the ice sheet will be asymmetric, with a relatively sharp onset and a decaying tail. This asymmetry is indeed seen in recent eruptions, but observations of eruption signals at EDC shows that the signals tend towards symmetry within a time (of order 1 kyr) that is relatively short compared to the >400 kyr of this study. For this reason, we chose the mathematical simplicity of assuming a Gaussian shape from the start. Our sensitivity studies (Fig. 3) where we changed the width of the initial peak between 1 and 5 years show that the exact width or shape of the initial peak does not materially affect the derived effective diffusion coefficients.
The text at L110-125 has been modified to make the difference in form (skewed versus Gaussian) between volcanic signals in the stratosphere versus an ice core clearer.

*In Figure 1, comparisons of the displayed scale bars showing 5 yr of ice accumulation with the observed sulfate peaks provide vivid illustrations of post-depositional ffects. However, the 5 yr span collapses onto a vertical line in the final 3 examples. I appreciate that the text gives further context, with the quoted 30 yr span for the 364 ka peak. However, I'd suggest modifying the figure caption or annotating each panel with the number of years that the 1m depth range represents.*

Figure 1 has been replaced following this suggestion.

*I found the theoretical development in sections 3.2 and 3.3 somewhat confusing. The standard convention in the modeling literature with which I am most familiar is to treat equations as portions of sentences, with appropriate punctuation (e.g. see Ng's 2021 paper). Instead, here you refer to the equations by number, and subsequently separate them out from the text. To me, this seems disjointed.*

We agree with the Reviewer's opinion and have modified sections 3.2 and 3.3 accordingly.

*In equations (6) and (7) you note that the effective dffusivity is expected to be a function of time. However, my understanding is that your model calculations in fact treat the dffusivity as constant through time is this correct?*

Yes, the Reviewer is correct. Effective diffusivity is treated as constant for each volcanic peak – it does not evolve with time. Equations 6 and 7 have been modified, following also the advice of Reviewer 2.

*It wasn't immediately obvious to me how equation (12) came about and why there is no explicit dependence on time. Indeed, a is really a rate, so I believe at/H is needed in the argument of the exponential to ensure dimensional correctness, and integration of (9) would produces this result following correction of a sign error.*

Yes, the equation should have at/H in the exponential. This has been corrected.

*On line 248 temperature and chemistry are mentioned as controlling variables. Perhaps grain size, or more generally, microstructure, should be mentioned as well.*

This suggestion has been adopted in revised manuscript (L810).

*On line 295 the very low eutectic temperature of sulfuric acid is used to justify the expectation that sulfate ions can be dissolved in liquid at EDC temperatures. However, in the paragraph beginning on line 155 you mention the Traversi et al. (2009) finding that appears to suggest that sulfate reacts with dust to presumably form a solid precipitate. Would it possibly be worth saying more here about the potential effects of chemical reactions between different impurity species?*

Yes, this is a good point. Mention of this possibility is now included (L876-879).

*The brief discussion in 5.3 begins by noting that the simplest version of Ng's (2021) model predicts much faster diffusion than is observed in the Holocene ice, which itself is faster than that observed in deeper regions. That the inferred diffusivity does not appear to depend on signal size would also seem to differ from Ng's (2021) model predictions. The proffered suggestion that Gibbs{Thomson diffusion efficiently reduces vein concentration gradients would appear to e ectively transform Ng's model to the Rempel et al. (2001) model, albeit only if vein density can evolve to enable signal translation. As the Barnes et al (2003) treatment relies upon ffects of grain-size evolution, it perhaps might contain some of the essential elements that these other two models lack. I'm not sure I follow the reasoning behind the nal sentence of this section. You've shown that the e ective di usion rate in the Holocene and early Pleistocene is both much slower than Ng's Gibbs{Thomson mechanism would predict and not systematically dependent on anomaly magnitude, so what makes you conclude that Ng's model correctly describes the controlling mechanism? I thought that I understood the Barnes-type model to depend on grain growth, but in the nal clause you say that the rate of grain growth isn't important. Please clarify.*

We thank the Reviewer for these insights thoughts and queries. Section 5.3 has been modified to make it clear that Ng's "Gibbs-Thomson" diffusion doesn't fit perfectly with our observations (it is too fast and we don't observe a rate dependence on signal size). At this stage we are not able to conclude on which mechanism(s) are operating but we do demonstrate that there must be a marked reduction in diffusion rate relatively early on.

*There's a typo in the title of the penultimate reference.*

Thanks.

Reply to Reviewer 2, Jeffrey L. Kavanaugh

We are grateful to Jeffrey L. Kavanaugh for his detailed and thoughtful review of our manuscript. We are confident that all concerns raised have been addressed in a revised manuscript. In particular, the description of our modelling approach is improved thanks to his queries and suggestions.

We do not reproduce the entire review here – sections of it are shown in italics. We note that some equations are not well-reproduced here. Our replies are given in regular font. Reviewer 2 also uploaded an annotated version of the manuscript, containing editorial suggestions that we are grateful for and have adopted with few exceptions.

1. Gaussian form

*…Their use here is more than justified*
*– but they remain just an approximation of the measured peak forms. (I'll also note that the description of stratospheric concentrations of sulphates following a major eruption (Lines 75–77) is decidedly non-Gaussian, being strongly asymmetric around the peak, and therefore it's reasonable for the reader to question whether sulphate concentrations in snow and ice are similarly asymmetric. Some additional discussion would help clarify this.*

A similar point was made by Reviewer 1 and our reply is repeated below. We note that the manuscript text states peaks are Gaussian in form "very shortly after deposition". We do not argue that the direct deposition of sulfate from the atmosphere is perfectly symmetric.

The reviewer is of course correct that the accumulation of sulfur in the stratosphere and its subsequent deposition to the ice sheet will be asymmetric, with a relatively sharp onset and a decaying tail. This asymmetry is indeed seen in recent eruptions, but observations of eruption signals at EDC shows that the signals tend towards symmetry within a time (of order 1 kyr) that is relatively short compared to the 400 kyr of this study. For this reason, we chose the mathematical simplicity of assuming an Gaussian shape from the start. Our sensitivity studies (Fig. 3) where we changed the width of the initial peak between 1 and 5 years show that the exact width or shape of the initial peak does not materially effect the derived effective diffusion coefficients.

*…it's unclear to me why FWHM is used to describe peak widths observed in the EPICA Dome C core throughout Section 2 (which describes the data), but FWTM is used throughout Section 3 (which describes the model). This isn't a major issue, to be sure (as the two are always related as FWTM/FWHM = 1.83), but it seems an unnecessary switch to make given that one value should be as easy to determine as the other from the data (but again, the data are presented in terms of FWHM, not FWTM). If FWTM is preferred, please include a brief explanation as to why.*

We agree, the use of FWHM and FWTM is confusing and not necessary. Section 2 has been modified to use the term peak width, which is close to FWTM that is used later in the analysis. Where peak width values are quoted for volcanic events in EDC (e.g., L131) these values can be found in the Supplementary Table. FWTM is preferred

(relative to FWHM)) for the model because it is closer to the 'width' metric of volcanic events in ice core sulfate identified by previous studies, e.g., Sigl et al., 2013, and therefore useful for comparison.

*Given that the objective here is to set up the numerical model, I recommend rearranging Section 3.2 somewhat, moving from expressing quantities in the time domain (FWTM/FWHM expressed in terms of years) to the spatial domain (FWTM/FWHM expressed in terms of distance) as quickly as possible, which could be accomplished by stating immediately stating that if the peak width is 3 years; moving Eq. 5 up to where it would become eq. 2; and then discussing relevant areas and fluxes (currently Eq. 2-4). (In my reading of this subsection, Eq. 2-4 needn't come before Eq. 5, but I might be missing something.)*

Section 3.2 has been modified but after some thought we decided it was better not to rearrange the equation order exactly as suggested by the Reviewer. This is because the text describes the steps taken in the order that we carry them out. This may not be the most efficient way to describe our approach but we hope it is helpful for readers to understand what we have done.

2. Description of the forward model

Reviewer 2 suggested re-writing equations 6 and 7 to use partial differentials and we have followed this advice (L422 and L427).

*Some clarification of the description of ice deformation at flow divides is also necessary. Lines 195-196 state "By assuming no lateral flow, the dynamics of the ice sheet consist only of one-dimension (vertical) flow, with ice layers thinning with increasing depth and pressure."*
*This misstates flow conditions in a couple of ways…*

We thank Reviewer 2 for his suggestions on how to improve the section on ice deformation. We have now used more appropriate terminology and removed the inaccurate statements (e.g., L427-430).

*It's important to also note that it is only because the same $\partial C/\partial x = 0$ and $\partial C/\partial y = 0$ conditions are met that diffusion (Eq. 6) can be treated as a 1-D problem here, rather than a 3-D one.*

This is now noted explicitly (L423).

*With respect to Eq. 6, the effective diffusion rate $D_{\mathrm{eff}}$ is expressed as a function of time (i.e., $D_{\mathrm{eff}}(t)$. However, given the description of the model in the text, it seems that the diQusion rate $D_{\mathrm{eff}}$ is held constant for each model run (with the best-fit diffusion rate for each sulfate peak determined independently from a set of 50 runs with log-spaced $D_{\mathrm{eff}}$ values). Is this correct?*

Yes, correct. The time dependence of Deff has been removed from Eq. 6 and 7. Text has been altered to include statement that Deff is time-invariant in our model (L425).

*If so, this would mean that the sulfate diffusion rate is determined by the time at which the snow fell, rather than by the length of time the snow is resident within the ice sheet – which*

*has implications for the interpretations regarding Gibbs-Thompson diffusion vs. slower processes discussed in Section 5.3.*

The 'time at which the snow fell' and the 'length of time the snow is resident in the ice sheet' are in effect the same value because the units of ice age ('time at which the snow fell') is thousands of years before present, if we understand the Reviewer correctly here.

We agree that using a constant effective diffusion coefficient negates the possibility of testing how a time-evolving diffusion coefficient (perhaps due to changes diffusion processes) impacts the outcome. In a future treatment we could attempt to derive the time varying diffusion coefficient under certain assumptions by starting diffusion from various depths or by adopting a finite difference calculation method, but that is beyond the scope of this paper. We have added a sentence to the summary to highlight this (L1141).

*Related to the discussion of the material derivative, it is not specified in the text whether the model is constructed in a Eulerian (i.e., fixed) coordinate system or a Lagrangian coordinate system (in which the coordinates track the deforming material). The framing of the equations suggests that a Eulerian coordinate system is used; this should be stated. (This is a relevant question because the sulfate peaks are advected downward through time.)*

A Eulerian coordinate system is used to generate a frame of reference relative to the centre of each peak.

*There are a few other concerns I have regarding the equations and phrasing in Section 3.3:*
*• The depth variable $z$ is defined (on Line 202) as the "height above the bed," but is subsequently referred to as "depth" (e.g. Line 205, which describes Eq. 9, and Line 207, which sets up Eq. 10). This is unnecessarily confusing to the reader, as thinking of $z$ as a depth-below-surface reverses the sign convention. It would be much clearer to refer to $z$ as "height above bed" throughout the text.*

Good point – manuscript has been modified.

*In Eq. 6-12, the spatial variable switches back and forth between $x$ and $z$. I suspect that $z$ refers to "depth within the ice sheet" and $x$ to "distance along the ice core," but didn't see this clarified in the text.*

Again, good point. Thanks. x is the distance along core relative to the centre of frame of reference (L420). Manuscript has been modified to clarify this.

*Eq. 8 is dimensionally incorrect: the left-hand side has units of m/m yr)\* = yr, whereas the right-hand side is dimensionless (units: m/m ); the equation is therefore a mathematical impossibility.*

We should have clarified that both the layer thickness lambda and the accumulation rate a are in m yr^-1. This is now done in the text.

*Equation 12 has a similar issue with dimensionality: the argument of an exponent must be dimensionless, whereas $-a/H$ has units of yr)\*.*

Yes, the equation should indeed have at/H in the exponential – thanks for spotting this typo.

*The equation defining the downward velocity field (Eq. 9) has issues with sign… Eq. 9 defines the ice velocity as being downward (i.e., negative). The negative sign ahead of the velocity term in Eq. 7 (as written in the manuscript) would therefore result in an upward (i.e. positive) velocity field. This is why the second term on the right-hand side of the material derivative equations must be positive in both 3-D and 1-D forms of the expressions for the material derivative.*

The sign error has been corrected.

**3. Significant figures**

We thank the Reviewer for his advice on significant figures, which we will followed throughout the manuscript. The only exception is the "median effective diffusion rate of sulfate ions of $2.4 \pm 1.7 \times 10^{-7}$ $m^2 yr^{-1}$ in Holocene ice" which appears in the Abstract and main text. Rounding both these values to $2 \pm 2 \times 10^{-7}$ $m^2 yr^{-1}$ would give the impression that Deff could be zero, which we are reluctant to do.

**4. Section 5.3 Implications**

*I'm not convinced this material needs to be presented separately from that presented in Section 5.1 ("Factors potentially influencing diQusion rate"), as the discussion as to whether and when the slower Barnes [2003]-type diQusion or the more rapid Gibbs-Thompson Ng [2023]-type diQusion might operate seems to fit well within that general topic.*

We respectfully disagree with the Reviewer's opinion here. We have modified the sub-heading of Section 5.3 to make the distinction between Section 5.1 and Section 5.3 clearer.

*I'm also not sure whether the study directly addresses whether Gibbs-Thompson diQusion might explain initial (high) sulfate diQusion rates, but not later (lower) rates of diQusion. This relates back to my earlier question regarding whether the eQective sulfate diQusion rate $D_{((}$ is held constant for a given sulfate peak during the model runs. If $D_{((}$ is held constant throughout each model run, the model does not directly answer the question: no "old" ice would have been modeled with high initial (Gibbs-Thompson) diffusion, followed by lower (Barnes-type) diffusion.*

The Reviewer is correct in stating that the diffusion rate does not change as a function of time for each individual peak. Each peak is modelled with a constant effective diffusion rate (Deff). However, Deff does not represent the instantaneous diffusion rate in ice of that peak's age but a time-weighted rate of diffusion over the entire history of the peak. By analysing each individual volcanic peak and assigning the 'best-fit' Deff for each one, we are able to ascertain that effective diffusion rate (and therefore diffusion rate) decreases with time. We are not able to quantify the change in instantaneous diffusion rate with time with our approach.

In a future treatment we could attempt to derive the time varying diffusion coefficient under certain assumptions by starting diffusion from various depths or by adopting a finite difference calculation method, but that is beyond the scope of this paper. But the reviewer is correct, at present we have not modelled any ice as having a high initial rate followed by a lower rate at some depth, although this is what we suspect is happening.

---

## Referee Report (RR1)

Although the authors have improved the quality of the manuscript with this revision, there remain some significant errors in the formulation and presentation of the mathematical model (Section 3.3) that I believe must be corrected before the manuscript can be considered for publication. Fortunately, the specific equations that have been used to calculate the time evolution of sulphate peaks appear to be correct, even if their mathematical derivations were flawed. My hope is that the derivations will be corrected so that mathematically-inclined readers will accept the model (and its results) as valid, and I will therefore focus on these corrections here. I believe that the results of this study are scientifically significant will prove to be valuable to ice core scientists following correction of the following errors in the model and/or its presentation:

(1) Eq. 8 is, as defined, physically nonsensical: a "layer thickness" cannot simply be assigned units of "m yr$^{-1}$", any more than one can accurately claim that "the distance between New York and Chicago is 790 miles per hour."

The correct form of this equation was derived by Nye [1963] and has been used by many others since (e.g., Cuffey and Paterson [2010]):

$$\frac{\lambda}{\lambda_0} = \frac{z}{H}$$

Here $\lambda_0$ is the original (ice-equivalent) layer thickness when precipitated on the surface of the ice sheet (located at constant height $H$ above the bed) and $\lambda$ is the thickness of the same layer when located at height $z$ above the bed some time later. In this expression, $\lambda$, $\lambda_0$, $z$, and $H$ all represent clearly defined spatial dimensions, and therefore all have units of length.

(2) Reading between the lines, it appears that the motivation for assigning units of "m yr$^{-1}$" to the layer thickness is to allow derivation of Eq. 9. through a simple algebraic manipulation of Eq. 8. This manipulation requires equating $v_z(z)$ to $z$ – a mathematical impossibility, given their different dimensions. Additionally, the origin of the negative sign in Eq. 9 is not immediately apparent, as it does not appear in Eq. 8.

Eq. 9 can be derived by starting with Nye [1963]'s assumption that "the vertical plastic strain-rate along any vertical line in the ice is uniform at any given instant," which can be expressed mathematically as

$$\dot{\varepsilon}_{zz} = -\frac{a_{ice}}{H}.$$

Here the negative strain rate indicates vertical compression of the ice, which is necessary to maintain a constant ice thickness $H$ despite the continuous addition of new ice to the surface (at rate $a_{ice}$).

To obtain the vertical ice velocity $v_z$ at height $z$ above the bed, we integrate the strain rate upwards from the base of the ice (where $v_x = v_y = v_z = 0$; this represents the second of the two assumptions that form the basis of Nye [1963]'s model):

$$v_z(z) = \int_0^z \dot{\varepsilon}_{zz}\, dz = \int_0^z -\frac{a_{ice}}{H}\, dz = -\frac{a_{ice}z}{H}$$

This method arrives at the correct expression for $v_z(z)$ without assigning physically nonsensical units to ice layers.

(3) In the revised manuscript, $x$ is defined as the "distance from the centre of the reference frame" (p.8, l.189-190). This is ambiguous: it defines neither specifically what "the reference frame" is, nor whether $x$ increases upwards (towards the ice surface) or downwards (towards the bed). From context, it can be inferred that the "centre of the reference frame" is "the center [or location of maximum concentration] of a given sulphate concentration peak" and that $x$ increases upwards, but these details should be stated clearly.

*As an aside, the "×" in Eq. 12 is unnecessary and can be omitted.

(4) Relatedly, in my review of the first version of this manuscript, I asked whether the model was framed in a Eulerian coordinate system (i.e., one that is spatially fixed) or a Lagrangian coordinate system (which follows an individual material parcel as it moves through time), as the distinction between $x$ and $z$ was not made in the initial manuscript. The authors responded that the model employed a Eulerian coordinate system.

This is not correct: the coordinate system tracks a given sulphate peak as it advects downward due to ice deformation and layer thinning, maintaining position $x(t) = 0$ even though the peak is continuously moving towards the base of the ice as time progresses. *The model is therefore cast in a Lagrangian coordinate system*.

This can also be seen by comparing the expression for $v_z(z)$ given in my point (2) above with the revised manuscript's Eq. 11. As $z$-space is defined (with $z = 0$ at the bed and $v = H$ at the ice surface), $z$ is positive wherever ice is present. So framed, vertical ice velocities are negative everywhere within the ice column, with all ice parcels moving downward towards the bed through decreasing $z$ values. Mathematically, ice layers thin in $z$-space because ice in the upper portions of a given layer have greater negative velocities than does ice in the lower portions of the layer.

In contrast, $x$-space is defined to have value $z = 0$ at the location of the sulphate peak, with negative $x$ values below the centre of the peak and positive $x$ values above the peak centre. As a result of this, the sign of the vertical velocity $v_x(x)$ differs for points above and below the peak's centre: it is negative for values $x > 0$ (i.e., downward for ice

above the peak), but positive for values $x < 0$ (i.e., upward for ice below the peak) – *even though the entire sulphate signal is being advected downwards in z-space*.

This isn't a problem, mathematically speaking: the flip in the velocity's sign is necessary to maintain thinning of the sulphate peak in $x$-space. Rather, I bring this up to again demonstrate that *the model is cast in a Lagrangian coordinate system*. Stating this explicitly would benefit the reader.

(5) It's not entirely accurate to state that "The Nye model ignores density changes" (p.9, l.203). Rather, the model requires that "ice-equivalents [be] used so as not to include snow and firn compaction in the strain-rate" (Nye [1963]).

Ice is commonly treated as incompressible in models, as this makes the deformation problem much more tractable. In the case of Nye [1963]'s model, this assumption allows the flow field to be uniquely defined with only the two simple assumptions given in points (1) and (2) above.

(6) There are some minor formatting errors related to Eq. 2, 5, 6, and 8 (for which text is included on the same line as the numbered equations), 9, and 12 (for which equation numbers appear on separate lines from their respective equations).

---

## Referee Report (RR2)

[referee-annotated manuscript omitted]

---

## Author Response (AR2)

**Reply to comments from Reviewer Jeffrey Kavanaugh.**
Rachael Rhodes et al.

We are grateful for Prof. Kavanaugh's assistance with the mathematical formulations included in our study. His input has greatly improved our manuscript.

Prof. Kavanaugh outlined 6 points where edits were required. We don't repeat his text here (the equations don't reproduce easily) but indicate where the requested changes have been made. Line numbers refer to the Tracked Changes version of manuscript.

[1] Equation 8 (L213) has been re-written as advised. Layer thickness units are all now metres.

[2] The correct derivation of our previous Eq. 9 (for vertical velocity) is now included in the manuscript. An additional intermediate equation provided by the Reviewer is now Eq. 9 (L217) and the new equation for $v(z)$ is Eq.10 (L220). Now it is clear to the reader where the negative sign comes from – thank you.
What was previously Eq. 11 has been deleted as it is not needed. This change doesn't seem to show up in Tracked Changes.

[3] The definition of distance $x$ has been improved (L193-194).

[4] Manuscript now clearly states that a Lagrangian coordinate system is used – apologies for lack of understanding here (L232) and thank you for persevering on this point.

[5] Manuscript has been edited to accurately reflect Nye model assumptions (L215).

[6] Formatting errors related to equations have been corrected.